# An ELISA Test Able to Predict the Development of Oral Cancer: The Significance of the Interplay between Steroid Receptors and the EGF Receptor for Early Diagnosis

**DOI:** 10.3390/diagnostics13122001

**Published:** 2023-06-08

**Authors:** Mariarosaria Boccellino, Alfredo De Rosa, Marina Di Domenico

**Affiliations:** 1Department of Precision Medicine, University of Campania “Luigi Vanvitelli”, 80138 Naples, Italy; 2Multidisciplinary Medical-Surgical Department, Odontostomatology Section, University of Campania “Luigi Vanvitelli”, 80138 Naples, Italy; 3Department of Biology, College of Science and Technology, Temple University, Philadelphia, PA 19140, USA

**Keywords:** oral cancer, androgen receptor, estrogen receptor, epidermal growth factors receptor, ELISA, cytological sampling

## Abstract

Oral disorders including non-homogeneous leukoplakia, erythroplakia, erosive lichen planus, and many others can potentially progress to oral squamous cell carcinoma (OSCC). Currently, the late diagnosis of OSCC contributes to high mortality rates, emphasizing the need for specific markers and early intervention. In this study, we present a novel, quick, sensitive, and non-invasive method for the early detection and screening of oral cancer, enabling the qualitative assessment of neoplastic forms even before the onset of symptoms. Our method directly examines the expression of oral cancer biomarkers, such as the epithelial growth factor receptor (EGFR), and steroid receptors, including the androgen receptor (AR) and the estrogen receptor (ER). The crosstalk between sexual hormones and the EGF receptor plays a crucial role in the progression of different types of cancers, including head and neck squamous cell carcinoma. To implement our method, we developed a kit box comprising nine wells or stations, each containing buffers, lysis systems, and dried/lyophilized antibodies stored at room temperature. The kit includes instruments for sample collection and a PVDF strip (Immobilon) with specific primary antibodies immobilized on it. These antibodies capture the target proteins from cytological samples. Additionally, complementary tools are provided to ensure efficient utilization and optimal test performance. The technique can be performed outside the laboratory, either “patient side” with an instant chemocolorimetric response or with a digital reader utilizing the enzyme-linked immunosorbent assay (ELISA) method.

## 1. Introduction

Oral cancer is a commonly occurring type of cancer that is becoming an increasing concern in various parts of the world. Each year, there are about 377,713 newly diagnosed cases of oral cancer worldwide (264,211 in males and 113,502 in females), resulting in a high death toll of 177,757 people (125,022 male; 52,735 female) [1]. In general, oral cancer encompasses various localizations, including the front two-thirds of the tongue, as well as the hard and soft palate, gums, floor of the mouth, buccal mucosa, and other parts of the mouth and oropharynx. The worldwide incidence of oral cavity cancers is estimated to be around 4 cases per 100,000 individuals. However, there is significant variation in this incidence rate across different regions, ranging from 0 to approximately 22 cases per 100,000 people [2]. Oral cancer is a prevalent disease that occurs more commonly in men and older individuals and is strongly associated with socioeconomic conditions. The most common type of oral cancer is oral squamous cell carcinoma (OSCC), accounting for approximately 95% of all oral cancers, and it usually develops in the oral mucosa, palate, and throat with squamous differentiation [3,4,5]. Despite significant progress in the diagnosis and treatment of OSCC, the 5-year survival rate for patients is still only 40–50% [6]. Major risk factors for oral cancer include tobacco smoking, alcohol consumption, and areca nut use, while human papillomavirus (HPV) infections are becoming more prevalent in North America and Europe, particularly among young people [7,8,9]. It is worth noting that more than a quarter of all oral cancer cases occur in people who do not smoke and only occasionally drink alcohol. Unfortunately, in the early stages of oral cancer, symptoms are often vague and difficult to identify, leading to a late diagnosis in many cases. This significantly reduces the potential survival rate from 80–90% to only 20–50% [10,11]. Therefore, improving early screening and diagnosis of oral cancer is crucial for enhancing the survival rate of patients.

The overexpression of the epidermal growth factor receptor (EGFR) correlates with the aggressive behavior of oral cancer. In fact, EGFR expression in oral cancer is a powerful independent prognostic indicator for overall survival (OS), disease-free survival (DFS), and a reliable predictor of the likelihood of local–regional (LR) relapse [12]. Targeting EGFR through monoclonal antibodies has shown positive outcomes, particularly in patients with recurrent or metastatic head and neck squamous cell carcinoma when other treatments have proven ineffective [13].

The signaling pathway mediated by steroid hormones plays a crucial role in the progression of OSCC. Therefore, targeting estrogen receptors (ER) in the treatment of ER-positive oral cancers, in combination with chemotherapy, may prove to be an effective strategy. Furthermore, studies have shown crosstalk between ER-mediated signals and the signaling pathway of EGFR, which has been implicated in the progression of various cancers, including head and neck squamous cell carcinoma [14,15,16]. However, despite significant progress in scientific research in recent years, no significant results have been reported yet in the diagnosis, prognosis, and treatment of oral cancer.

The objective of this research was to develop a novel, quick, efficient, and non-invasive method for the early detection and screening of oral cancer, enabling the identification of malignant forms during the pre-symptomatic phase. In particular, we focused on the expression of specific markers of oral cancer, namely the epidermal growth factor receptor (EGFR) and steroid receptors such as the androgen receptor (AR) and the estrogen receptor (ER). The device is constructed by combining off-the-shelf and custom-made materials. The technique is carried out “patient side” in non-laboratory settings with an instant chemocolorimetric response or using a digital reader based on an ELISA method. The method is described in a patent application titled “In vitro screening method and kit for early diagnosis of oral cavity tumours,” by the inventors Marina Di Domenico, Mariarosaria Boccellino, and Alfredo De Rosa (Patent-Italy-N.IT102018000004137; WIPO PCT/140315 International Patent System).

## 2. Materials and Methods

### 2.1. Antibodies

Antibody anti-AR Ab N-20 sc 816 (Santa Cruz Biotechnologies Inc., Santa Cruz, CA, USA); anti-ER antibody sc 543 (Santa Cruz Biotechnologies Inc., Santa Cruz, CA, USA); anti-EGFR antibody sc-03 g (Santa Cruz Biotechnologies Inc., Santa Cruz, CA, USA); and actin antibody were purchased from GeneTex (Irvine, CA, USA). The PVDF strip was from ThermoFisher Scientific (Monza MB, Italy).

### 2.2. Cell Culture

KB cells, a type of epidermal carcinoma of the oral cavity, were obtained from the National Centre for Cell Science. The cells were cultured in Dulbecco’s modified Eagle medium supplemented with 10% FBS (fetal bovine serum) and 1% streptomycin–gentamicin antibiotics. They were then incubated at 37 °C in a humidified atmosphere with 5% CO_2_. The cells were maintained in the exponential growth phase and were subcultured using 0.25% trypsin [17,18].

### 2.3. Western Blot

The proteins were subjected to gel electrophoresis, followed by transfer and visualization using standard techniques, as previously described [19,20]. Briefly, KB cells were lysed in RIPA lysis buffer (150 mM NaCl, 1% NP-40, 50 mM Tris pH 8.0, 0.5% sodium deoxycholate, 0.1% SDS) containing a protease inhibitor cocktail (Sigma-Aldrich, St. Louis, MO, USA) at 4 °C for 1 h. After centrifugation at 14,000× *g* for 15 min at 4 °C to separate the cell debris from the proteins, the proteins were separated using 10% SDS-PAGE gels and transferred onto a nitrocellulose membrane. The membrane was then incubated with various antibodies overnight at 4 °C [21,22]. Actin was used as a control to normalize the sample loading. The proteins were visualized using peroxidase-labeled protein A (200 ng/mL) and detected using ECL Plus detection reagents (Amersham, GE Healthcare Bio-Sciences Corp., Piscataway, NJ, USA). 

### 2.4. PVDF Strip Preparation Protocol 

The steps to prepare the PVDF strip equipped with the primary antibodies of interest were conducted in a sequence. First, the PVDF strip was hydrated with methanol for five minutes, followed by two washes with PBS for another five minutes. The strip was then incubated with protein A at a concentration of 10 µg/mL in PBS for one hour and washed twice with PBS. After washing, it was blocked with a 3% BSA solution in PBS for an hour and rinsed three times with PBS for five minutes each. Finally, the PVDF strips were incubated separately with a solution containing rabbit antibodies against AR, ER, EGFR, or actin at a concentration of 3 µg/mL overnight at 4 °C with gentle shaking. Strips were then washed three times with PBS for 5 min each, followed by two washes with PBS TEA 0.2 M. After that, the strips were incubated with 25 mM DMP in TEA HCl 0.2 M at pH 8.2, followed by a solution containing TEA at 0.2 M and 20 mM ethanolamine. The strips were then washed with PBS for five minutes twice. Finally, the strips were stored in 0.02% NaN_3_ in PBS. To fit the device support for performing the ELISA analysis, the strips were cut to a width of approximately 0.4 cm.

### 2.5. Device Design

The device employed dried or lyophilized antibodies that remain stable at room temperature. These antibodies were solubilized in a suitable buffer solution at the time of use. The device was assembled using commercially available or custom-made semi-finished products. It included an instrument (Cytobrush) for collecting a biological sample from the oral mucosa. Additionally, it comprised all the necessary components for analyzing the collected sample using the ELISA method and verifying the presence of biomarkers. The product has a straightforward and easy-to-use method.

### 2.6. Description of the Device

The kit box was organized into rows of nine wells or stations. Each well contained buffers, lysis systems, and detection systems that included dried and/or lyophilized antibodies, which can be stored at room temperature (as shown in Figure 1). The kit box also included instruments for collecting biological samples and a PVDF strip (Immobilon) on which the specific primary antibodies were adhered. Additionally, the kit provided tools that facilitated efficient utilization and performance of tests.

### 2.7. Test Procedure

(1) The substances present in the wells/station were in a dried or lyophilized form, and they needed to be dissolved in a suitable buffer before using them (Figure 1).

(2) The collection of cells from the oral mucosa was carried out with the appropriate instruments (Figure 2A). 

(3) The biological sample was mixed with the lysis buffer in well/station number zero for 15 min (Figure 2A). 

(4) The PVDF membrane was immersed for 8 min in well/station number one for antigen recognition by the primary antibodies attached to the PVDF strip (Figure 2B). When proteins were present in the sample being tested, the primary antibodies on the strip formed immune complexes with the antigens, which were indicative of the presence of proteins in the sample (Figure 2C).

(5) After immune complex formation, the PVDF strip was transferred to well/station number two with a solution containing primary monoclonal antibodies that specifically bind to immune complexes. The strip was immersed in this solution for 5 min (Figure 2D).

(6) The PVDF strip holding the immune complexes was washed in wells/stations number three and four using a T-PBS-buffer solution. This washing step was necessary to remove any non-specifically attached proteins that may have been present in the immune complex (Figure 2E,F).

(7) In well/station number five, the PVDF strip was exposed to a solution containing secondary antibodies conjugated with an enzymatic detection system (alkaline phosphatase) for 5 min (Figure 2G).

(8) The PVDF strip was washed in wells containing T-PBS buffer at stations number six and seven to remove any excess secondary antibodies (Figure 2H,I).

(9) In well/station number eight, the PVDF strip was exposed to a solution of the substrate (BCIP/NBT) for 4 min to initiate the colorimetric reaction (Figure 2J).

The colorimetric assay can be performed at a temperature range of 20 °C to 30 °C. This temperature range is ideal because the enzymatic activity of alkaline phosphatase or peroxidase linked to the secondary antibody is highest within this range, which results in optimal signal amplification. 

## 3. Results

This method directly assesses oral cancer biomarkers using the ELISA test to determine the levels of EGF, androgen (AR), and estrogen (ER) receptors. It allows for the qualitative identification of these receptors, which are indicative of oral cancer, using a cytological sampling technique. The test involves immobilizing a series of antibodies specific to the target proteins onto a membrane. After the membrane is immobilized with the specific antibodies, it is then exposed to a solution containing a sample of cells. If the desired marker proteins are present in the sample, they will be captured by the antibodies on the membrane. The membrane is then washed and incubated with a solution containing secondary antibodies conjugated with alkaline phosphatase. This forms a sandwich structure that can be detected using a chemocolorimetric technique. The membrane is subsequently placed in a chemocolorimetric detection station, where the presence of the protein–antibody complex causes a pink/purple band to appear, indicating the presence of the target protein.

To evaluate the sensitivity of the test, we conducted in vitro investigations to validate the method. The most common and widely occurring category of oral cancer is epithelial tumors, followed by salivary gland tumors. Within the epithelial tumor category, squamous cell carcinoma is the most prevalent. Moreover, the oral mucosa is the most frequent location for oral cancer, followed by the tongue [23,24]. Therefore, for the validation of the designed tool, the cellular extracts used in the laboratory were obtained from KB cells derived from human oral squamous cell carcinoma.

As a first approach, we evaluated the expression of the three receptors in KB cells in vitro using Western blot analysis. To optimize the incubation times, we conducted experiments to determine the optimal cell lysis time. For this purpose, KB cells were incubated in lysis buffer to increase the durations up to 1 h (0, 15, 30, 60 min), and the extracted proteins were analyzed via Western blot using specific primary antibodies against EGFR, AR, and ER (Figure 3). The highest protein concentration (0.55 µg/µL) was detected after a lysis time of 15 min. As shown in Figure 3, the expression of the three target proteins was detected in all analyzed samples. 

To confirm the results obtained by Western blot, we performed ELISA assays on the PVDF strips of the device to detect the same antigens derived from the protein lysates of the KB cells.

Figure 4 shows the results regarding the expression of the three proteins (EGFR, AR, and ER) on the PVDF strips prepared as described in the Section 2. Each antibody was immobilized on a separate PVDF strip. Subsequently, the strips were immersed in a 400 µL solution containing a cell lysate with concentrations ranging from 100 to 5 µg/µL (Figure 4). In this case, a highly concentrated pink or purple color was observed at the end of the ELISA assay.

Successive experiments conducted with decreasing protein concentrations down to 1 µg/µL still yielded positive results. Furthermore, additional experiments performed with protein concentrations below 50 ng demonstrated that the method is capable of detecting concentrations as low as 10 ng (data not shown).

## 4. Discussion

Oral squamous cell carcinoma constitutes approximately 90% of all oral cancers and represents one of the most prevalent malignancies worldwide. Metastasis is the primary cause of death in this cancer type. The molecular pathogenesis of head and neck squamous cell carcinoma is complex and multifactorial, involving various cytogenetic alterations that drive a progressive transition from normal mucosa to dysplasia, carcinoma in situ, and advanced cancer stages [25,26,27,28]. 

Early-stage OSCC is challenging to detect, resulting in a majority of patients being diagnosed at advanced stages [29,30]. Hence, the development of novel screening and early diagnosis approaches is crucial for reducing OSCC-related morbidity and mortality rates.

Several studies have investigated the role of EGFR in the pathogenesis of oral carcinoma. Around 80% of oral squamous cell carcinomas exhibit EGFR overexpression, which promotes the proliferation and differentiation of keratinocytes [31]. The overactivation of the EGFR pathway is recognized as an etiological factor in human cancer, contributing to cancer development, metastasis, and chemotherapy resistance. EGFR overexpression in oral cancer correlates with a malignant phenotype, suppression of apoptosis, and increased metastatic potential [32].

ERs are expressed not only in the human genitalia but also in various organs. Multiple studies have suggested that ERs play a crucial role in carcinogenesis by promoting cancer progression, including proliferation, invasion, and chemoresistance, in various cancers, such as OSCC and colon cancer. These effects are distinct from those observed in sex-steroid-dependent organs. Moreover, ERα activation has been implicated in enhancing the invasiveness of prostate cancer cells, implying its potential contribution to tumor cell motility through the epithelial–mesenchymal transition (EMT) process [33,34,35].

Cross-signaling between the ER and EGFR pathways has been observed in various tumor types, including esophageal cancer, indicating cooperative interactions between these receptors in tumor progression [15,36]. The androgen receptor (AR) is emerging as a promising tumor target for improved diagnostics and therapeutic options due to its significant role in the etiology and progression of different cancer types. Additionally, AR expression is crucial for tumor cell growth in OSCC [37,38,39,40].

We have previously observed crosstalk between EGFR and extranuclear steroid receptors (AR and ER) in tumor cells (MCF-7 and LNCaP). In these cells, the AR/ER/Src complex is necessary for the action of EGF. Specifically, EGF stimulates ER phosphorylation and promotes the formation of a complex involving EGFR, AR/ER, and the activation of the Src kinase strongly influences EGFR phosphorylation, leading to the activation of the EGF-dependent signaling pathway [41].

In this expanding and intriguing context, we hypothesize that the simultaneous expression of the three receptors, AR, ER, and EGFR, is associated with a phenotypic transition of cells from epithelial to mesenchymal, driven by alterations in cellular metabolism that underlie the epithelial–mesenchymal transition (EMT). These changes ultimately contribute to malignant transformation. This screening method not only facilitates the detection of neoplastic forms in the pre-symptomatic early stages but also allows for timely interventions, potentially leading to improved patient outcomes. Early detection of oral cancer is crucial to implement effective treatment strategies and enhance survival rates. By identifying oral cancer at an early stage, treatment options can be less invasive, potentially sparing patients from extensive surgical procedures or aggressive therapies. Additionally, early detection enables targeted interventions that specifically target the molecular pathways involved in tumor progression, offering personalized treatment strategies for enhanced therapeutic efficacy. Furthermore, this screening method has the potential to reduce the burden on healthcare systems by minimizing the need for extensive diagnostic procedures and treatments for advanced-stage oral cancer cases. The method’s non-invasiveness, rapidity, sensitivity, specificity, and ease of use make it a promising approach for widespread adoption and implementation in both clinical and community settings. The method is based on the ELISA technique showing a chemocolorimetric result for membranes with adsorbed antibodies for the three proteins (EGFR, AR, and ER) found in the oral sample. The method can be assembled as a kit, which makes it easily transportable and usable in the field “patient side” as it does not need any particular equipment. The kit utilizes stable reagents that can be stored at room temperature, and it offers a rapid, sensitive, specific, and non-invasive testing method.

## 5. Conclusions

The current invention is designed to be an efficient diagnostic aid for early oral cancer detection without the need for laboratory equipment, and it provides immediate results upon testing. Among the strengths of the product, it should be noted that it has the particularity of allowing a non-invasive, particularly reliable, very fast, and economical investigation. It effectively responds to the need for prevention, which, with reference to carcinomas of the oral cavity, is now a need given their highly aggressive pathology and poor prognosis. The only cure is thus achieved with early diagnosis. In vivo screening is required to validate the in vitro method.

## 6. Patents

The method is described in a patent application titled “In vitro screening method and kit for early diagnosis of oral cavity tumours” by the inventors Marina Di Domenico, Mariarosaria Boccellino, and Alfredo De Rosa (Patent-Italy-N.IT102018000004137; WIPO PCT/140315 International Patent System).

## Figures and Tables

**Figure 1 diagnostics-13-02001-f001:**
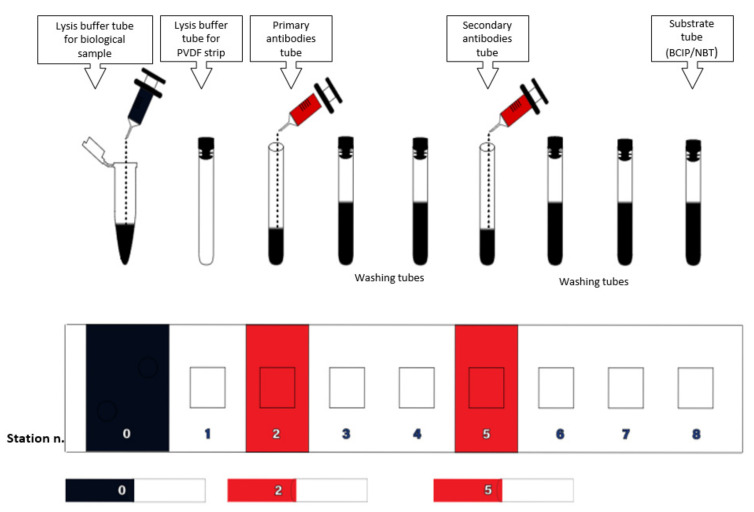
Scheme of the kit with 9 wells/stations. Upon opening the kit, the dried reagents in stations 0, 2, and 5 were dissolved in a suitable buffer. Station 0 contained the lysis buffer for biological sample lysis. Station 2 contained a solution of primary monoclonal antibodies (anti-EGFR, -ER, -AR), and station 5 contained a solution of secondary antibodies with an enzymatic detection system.

**Figure 2 diagnostics-13-02001-f002:**
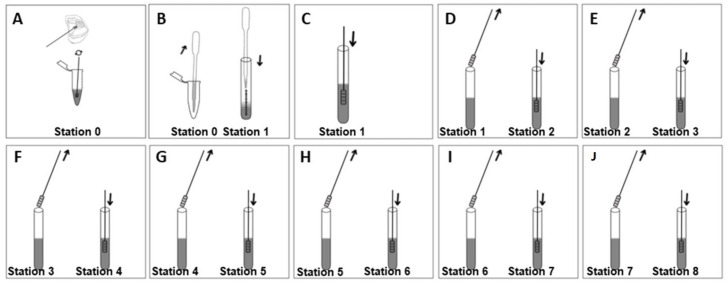
Device test procedure. In station 0, the biological sample underwent lysis. In station 1, the antigens were recognized by the specific antibodies attached to the PVDF strip. In station 2, the immune complex was recognized by the primary antibodies in the solution. Stations 3 and 4 were used for washing. In station 5, incubation occurred with the secondary antibodies conjugated with an enzymatic detection system. Stations 6 and 7 were used for additional washings. Finally, in station 8, incubation took place with the substrate (BCIP/NBT).

**Figure 3 diagnostics-13-02001-f003:**
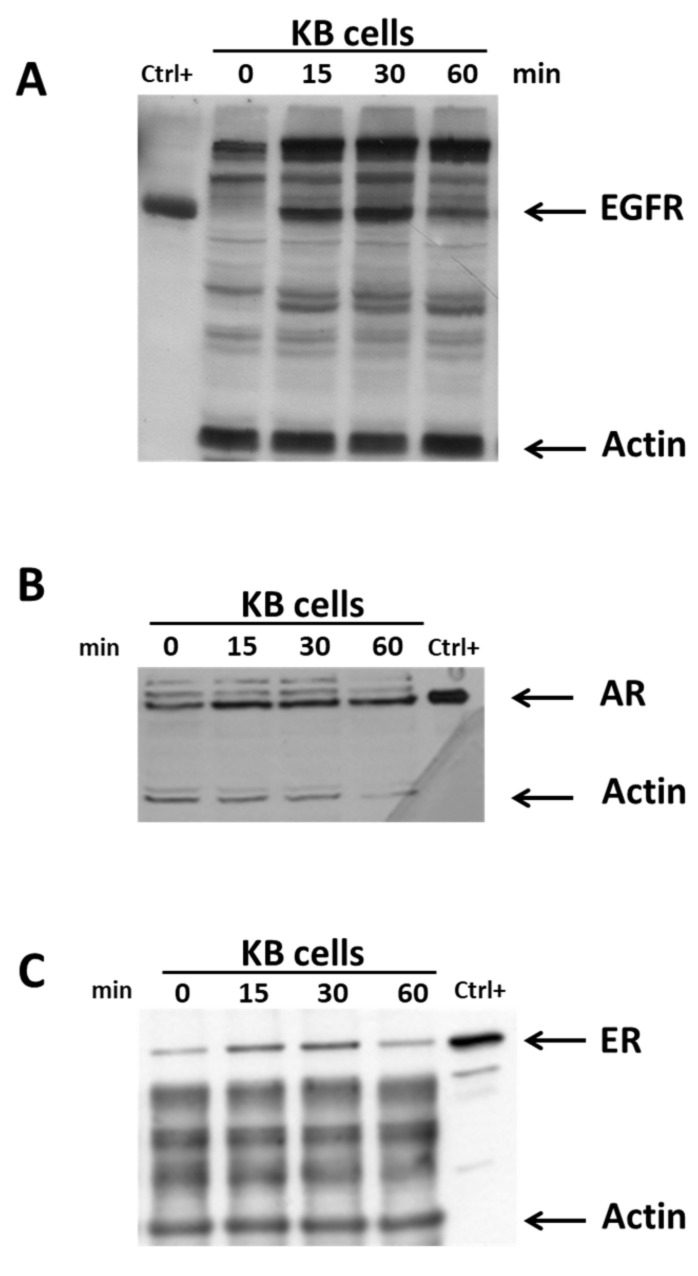
Western blot analysis of EGFR (**A**), AR (**B**), and ER (**C**) expression. KB cells were cultured for 48 h and then lysed to prepare protein extracts. Electroblot on the nitrocellulose filter was incubated with EGFR- or AR- or ER- or actin specific antibodies and detected by chemiluminescence. Data are representative of three independent experiments for each antibody.

**Figure 4 diagnostics-13-02001-f004:**
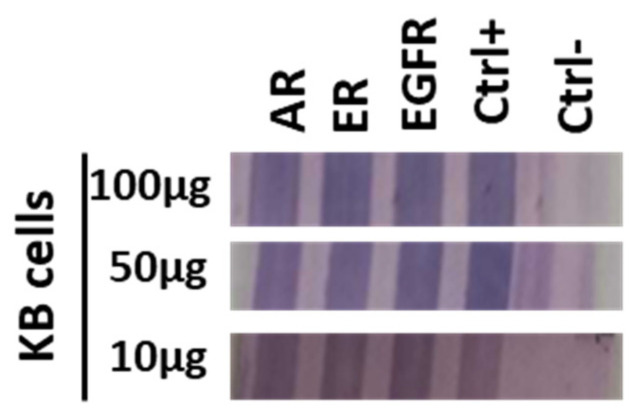
ELISA assay. The strips were immersed in a solution containing a cell lysate (range: 100–5 µg/µL). A very intense pink/purple colorimetric effect was obtained.

## Data Availability

The data presented in this study are available on request from the corresponding author.

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
