# Peer review of "An ELISA Test Able to Predict the Development of Oral Cancer: The Significance of the Interplay between Steroid Receptors and the EGF Receptor for Early Diagnosis"

_diagnostics, 2023, doi:10.3390/diagnostics13122001_

Round 1

Reviewer 1 Report

Even though this study is an interesting and important study it lacks the clarity. It is very confusing and is extremely difficult to follow the manuscript. Methodology section needs clarity and must include all the necessary details. Institutional review board clearance must be included. Discussion is superficial without any meaningful comparison. It is not clear how the results are going to help the claim. Authors have missed some important and recent articles. Language needs editing.

Author Response

Point-by-point answers

Even though this study is an interesting and important study it lacks the clarity. It is very confusing and is extremely difficult to follow the manuscript. Methodology section needs clarity and must include all the necessary details.

Response:

We appreciate your feedback on the clarity and inclusion of necessary detail in the methodology section. In the revised version of the manuscript, we have carefully revised the methodology, incorporating more complete and explicit information to ensure clarity and completeness as suggested.

Institutional review board clearance must be included.

Response:

Regarding the statement that 'Institutional Review Board clearance must be included', I would like to clarify that my research, was conducted in vitro using KB cells, which are a type of epidermal carcinoma of the oral cavity. As the experimentation did not involve the use of humans or animals, and solely utilized cell cultures, no authorization from the ethics committee was required.

Discussion is superficial without any meaningful comparison. It is not clear how the results are going to help the claim.

Response:

We thank the reviewer for this suggestion, in the revised version of the manuscript, we have provided the discussion section with a more in-depth analysis of the findings and their implications as follows:

This screening method not only facilitates the detection of neoplastic forms in the pre-symptomatic early stages but also allows for timely interventions, potentially leading to improved patient outcomes. Early detection of oral cancer is crucial in order to implement effective treatment strategies and enhance survival rates. By identifying oral cancer at an early stage, treatment options can be less invasive, potentially spar-ing patients from extensive surgical procedures or aggressive therapies. Additionally, early detection enables targeted interventions that specifically target the molecular pathways involved in tumor progression, offering personalized treatment strategies for enhanced therapeutic efficacy. Furthermore, this screening method has the potential to reduce the burden on healthcare systems by minimizing the need for extensive diagnostic procedures and treatments for advanced-stage oral cancer cases. The method's non-invasiveness, rapidity, sensitivity, specificity, and ease of use make it a promising approach for widespread adoption and implementation in both clinical and community settings”.

Authors have missed some important and recent articles.

Response:

We thank the reviewer for this advice, we have added some recent and important articles in the revised version of our manuscript, as follow:

-Piao, Y.; Jung, S.N.; Lim, M.A.; Oh, C.; Jin, Y.L.; Kim, H.J.; Nguyen, Q.K.; Chang, J.W.; Won, H.R.; Koo, B.S. A circulating microRNA panel as a novel dynamic monitor for oral squamous cell carcinoma. Sci Rep. 2023, 3;13(1), 2000.

-Badwelan, M.; Muaddi, H.; Ahmed, A.; Lee, K.T.; Tran, S.D. Oral Squamous Cell Carcinoma and Concomitant Primary Tumors, What Do We Know? A Review of the Literature. Curr. Oncol. 2023, 30, 3721-3734.

Language needs editing.

Response:

The language has been edited and revised as suggested.

Reviewer 2 Report

The authors described "An ELISA test useful to predict the development of oral cancer: significance of the interplay between steroid receptors and the EGF receptor for the early diagnosis". This topic should be important and informative for potential readers because early-stage OSCC is difficult to detect. This simultaneous ELISA test may be useful. However, I have some questions.

1. In abstract, they should change the contents because introduction part was too long, also methods and results should be added.

2. Is this method really useful for the patients? How was a pilot test in clinical practice? 

3. How is the other marker (e.g. SCC)? Please add the reason for including and excluding the marker in Introduction or Discussion.

Author Response

Point-by-point answers

The authors described "An ELISA test useful to predict the development of oral cancer: significance of the interplay between steroid receptors and the EGF receptor for the early diagnosis". This topic should be important and informative for potential readers because early-stage OSCC is difficult to detect. This simultaneous ELISA test may be useful. However, I have some questions.

Response:

We thank the reviewer for his comments.

  1. In abstract, they should change the contents because introduction part was too long, also methods and results should be added.

Response:

We thank the reviewer for this suggestion, we have rewritten the abstract including methods and results as suggested. Here is the revised abstract:

“Oral disorders including non-homogeneous leukoplakia, erythroplakia, erosive lichen planus, and many others, can potentially progress to oral squamous cell carcinoma (OSCC). Currently, the late diagnosis of OSCC contributes to high mortality rates, emphasizing the need for specific markers and early intervention. In this study, we present a novel, quick, sensitive, and non-invasive method for the early detection and sceening of oral cancer, enabling the qualitative assessment of neoplastic forms even before the onset of symptoms. Our method directly examines the expression of oral cancer biomarkers, such as the epithelial growth factor receptor (EGFR), and steroid receptors, including the androgen receptor (AR) and the estrogen receptor (ER). The cross talk be-tween sexual hormones and EGF receptor play a crucial role in the progression of different types of cancers including head and neck squamous cell carcinoma. To implement our method, we have developed a kit box comprising nine wells or stations, each containing buffers, lysis systems, and dried/lyophilized antibodies stored at room temperature. The kit includes instruments for sample collection and a PVDF strip (Immobilon) with specific primary antibodies immobilized on it. These antibodies capture the target proteins from cytological samples. Additionally, complementary tools are provided to ensure the efficient utilization and optimal test performance. The technique can be performed outside the laboratory, either at the “patient side” with an instant chemocolorimetric response or with a digital reader utilizing the enzyme-linked immunosorbent assay (ELISA) method”.

  1. Is this method really useful for the patients? How was a pilot test in clinical practice?

Response:

This method has undergone in vitro testing, and we are currently in the process of planning an in vivo validation. It is crucial to acknowledge that further clinical evaluation is required to determine the effectiveness and utility of the method for patients. While the initial results and proposed approach show promise for early detection and screening of oral cancer, conducting pilot trials in clinical practice is essential to assess its performance in real-world scenarios. By conducting pilot trials in clinical practice, we aim to evaluate the method's feasibility, accuracy, and acceptability in a clinical setting. These trials will provide valuable data on the method's performance, potential benefits, and any challenges that may arise during implementation. It will also enable us to assess the impact of the method on patient care and its practicality for healthcare professionals.

  1. How is the other marker (e.g. SCC)? Please add the reason for including and excluding the marker in Introduction or Discussion.

Response:

The focus of our research was to develop a novel, quick, efficient, and non-invasive method for the early detection and screening of oral cancer during the pre-symptomatic phase. In line with this objective, we specifically targeted markers that have been widely associated with oral cancer, namely the epithelial growth factor receptor (EGFR) and steroid receptors such as the androgen receptor (AR) and the estrogen receptor (ER). These markers have shown significant relevance in the pathogenesis and progression of oral cancer. However, the exclusion of the SCC marker does not diminish its significance in oral cancer diagnosis. Future studies could explore its potential in combination with other markers or in specific clinical contexts where its utility has been demonstrated.

Reviewer 3 Report

The manuscript presents a novel method of early oral cancer detection. Here, the author used the OSCC biomarker, such as EGFR, and steroidal biomarkers, such as ER and AR, and devised the ELISA method for early oral cancer detection. This manuscript is well-written, and the methodology and results support the study's novelty. Some minor comments need to be addressed: 

1.     The author explained the device in schematic Figure 1.  It will be possible to provide the original image of the device with all components and about it in the legend.

2.     Did this method apply to some clinical pilot studies?

3.     Redesign figures 1 and 2 with legends having two or three sentences. 

Author Response

Point-by-point answers

The manuscript presents a novel method of early oral cancer detection. Here, the author used the OSCC biomarker, such as EGFR, and steroidal biomarkers, such as ER and AR, and devised the ELISA method for early oral cancer detection. This manuscript is well-written, and the methodology and results support the study's novelty. Some minor comments need to be addressed:

Response:

We thank the reviewer for the appreciation to our work.

  1. The author explained the device in schematic Figure 1. It will be possible to provide the original image of the device with all components and about it in the legend.

Response:

We have added more information about the components in Figure 1.

  1. Did this method apply to some clinical pilot studies?

Response:

As already answered to reviewer 2, until now, the method has not been applied to any clinical studies. Currently, we are planning the next phase of in vivo validation to assess the efficacy and utility of the method in a real clinical setting. While the initial results and proposed approach show promise for early detection and screening of oral cancer, conducting pilot trials in clinical practice is essential to assess its performance in real-world scenarios. By conducting pilot trials in clinical practice, we aim to evaluate the method's feasibility, accuracy, and acceptability in a clinical setting. These trials will provide valuable data on the method's performance, potential benefits, and any challenges that may arise during implementation. It will also enable us to assess the impact of the method on patient care and its practicality for healthcare professionals.

  1. Redesign figures 1 and 2 with legends having two or three sentences.

Response:

We thank the reviewer for this suggestion. We have added more detail to the legends in figures 1 and 2 as follows:

Figure 1. Scheme of the kit with 9 wells/stations. Upon opening the kit, the dried reagents in stations 0, 2, and 5 are dissolved in a suitable buffer. Station 0 contains the lysis buffer for biological sample lysis. Station 2 contains a solution of primary monoclonal antibodies (anti-EGFR, -ER, -AR), and station 5 contains a solution of secondary antibodies with an enzymatic detection system.

Figure 2. Device test procedure. In station 0, the biological sample undergoes lysis. In station 1, the antigens are recognized by the specific antibodies attached to the PVDF strip. In station 2, the immune complex is recognized by the primary antibodies in solution. Stations 3 and 4 are used for washing. In station 5, incubation occurs with the secondary antibodies conjugated with an enzymatic detection system. Stations 6 and 7 are for additional washings. Finally, in station 8, incubation takes place with the substrate (BCIP/NBT).

Reviewer 4 Report

Authors of this manuscript focused on the expression of specific markers of oral cancer, the epithelial growth factor receptor (EGFR), the androgen receptor (AR) and the estrogen receptor (ER) by using an ELISA method. The device is built by assembling commercially available and custom made semi-finished products. The method is performed in environments outside the laboratory, "patient side" with immediate chemo-colorimetric response or with digital reader by using an ELISA method.

The invention is interesting and its use in outside laboratory conditions could be interesting. The feasibility is confirmed by in vitro assays. Subsequent in vitro tests will follow, which will also allow to determine the range of application, which could be interesting also in follow up after successful therapy.

Author Response

Point-by-point answers

Authors of this manuscript focused on the expression of specific markers of oral cancer, the epithelial growth factor receptor (EGFR), the androgen receptor (AR) and the estrogen receptor (ER) by using an ELISA method. The device is built by assembling commercially available and custom made semi-finished products. The method is performed in environments outside the laboratory, "patient side" with immediate chemo-colorimetric response or with digital reader by using an ELISA method.

The invention is interesting and its use in outside laboratory conditions could be interesting. The feasibility is confirmed by in vitro assays. Subsequent in vitro tests will follow, which will also allow to determine the range of application, which could be interesting also in follow up after successful therapy.

Response:

We appreciate the reviewer's recognition of our work. As previously mentioned in our response to reviewer 2, our method has undergone in vitro testing, and we are currently in the process of planning an in vivo validation. It is important to emphasize that further clinical evaluation is necessary to determine the effectiveness and utility of the method for patients. While the initial results and proposed approach show promise for early detection and screening of oral cancer, conducting pilot trials in clinical practice is crucial to evaluate its performance in real-world scenarios. By conducting pilot trials in clinical practice, our goal is to assess the feasibility, accuracy, and acceptability of the method in a clinical setting. These trials will provide valuable data on the method's performance, potential benefits, and any challenges that may arise during implementation. Additionally, we will evaluate the impact of the method on patient care and its practicality for healthcare professionals. We are committed to rigorously testing and validating the method through comprehensive clinical evaluations. The results obtained from these pilot trials will contribute to a better understanding of the method's clinical applicability and its potential to enhance early detection and treatment outcomes for individuals with oral cancer.

Round 2

Reviewer 1 Report

Authors have not addressed the concerns satisfactorily

Reviewer 2 Report

The authors revised the manuscript precisely. Thank you for this opportunity.